# Self-Guided Semantic Alignment for Text Supervised Segmentation

## Abstract

Learning to segment images by purely relying on the image-text alignment from web data can lead to sub-optimal performance due to noise in the training data. The noise comes from the samples where the associated text does not or only partially describes the image's visual content. Instead, this work proposes a novel loss function termed SimCon, which compares an image jointly to images and texts while accounting for intra-modal similarities to determine the appropriate set of semantic positives. Further, using multiple views of the image (created synthetically) and combining the SimCon loss with it makes the training more robust. This version of the loss is termed MV-SimCon. The empirical results demonstrate that using the proposed loss function leads to consistent improvements on zero-shot, text supervised semantic segmentation and outperforms state-of-the-art by +3.0%, +3.3% and +6.9% on PASCAL VOC, PASCAL Context and MSCOCO, respectively. With test time augmentations, we set a new record by improving these results further to 58.7%, 26.6%, and 33.3% on PASCAL VOC, PASCAL Context, and MSCOCO, respectively. In addition, using the proposed loss function leads to robust training and faster convergence.

## 1 Introduction

Using data from the web for training visual and language models has been effective in learning useful visual representations. Training the visual and textual encoders jointly via projecting images and texts to the same learned embedding space allows their direct comparison, which helped to develop open-set and zero-shot classification models Radford et al. (2021); Jia et al. (2021); Alayrac et al. (2022); Lu et al. (2021); Chen et al. (2022); Yu et al. (2022); Singh et al. (2022); Yao et al. (2021); Zhai et al. (2022b); Pham et al. (2021); Radenovic et al. (2023). This success has prompted researchers to investigate using web data for learning object-level representations for tasks involving dense predictions such as semantic segmentation Xu et al. (2022a), without training on any dense supervision. Whether the focus is on learning global or object-level representations, these approaches rely on aligning the images and the co-occurring text for supervision.

The cross-modal alignment for zero-shot image classification or segmentation models is performed via contrastive learning using InfoNCE Hadsell et al. (2006); van den Oord et al. (2018) loss function, which maximizes the mutual information between the image and its matched text. The loss is computed by using each sample in the batch as an anchor. In the embedding space, for an anchor image (text), the embedding vector of the corresponding text (image) is pulled closer. In contrast, the embedding vectors of texts (images) from other samples are pushed apart. While this objective has been useful Radford et al. (2021); Jia et al. (2021); Xu et al. (2022a), it is prone to noise in the training data, which is typical for samples from the web. Often the text associated with an image does not describe its visual content, it may miss the information about objects in the background, or it could be ambiguous. As shown in Fig. 1, the caption associated with the first image does not describe the visual content, whereas the caption for the second image describes the visual content of both the first and the second image. Using the InfoNCE loss function for these samples will push apart the embedding vectors of the first image and the second text, potentially leading to sub-optimal representations.

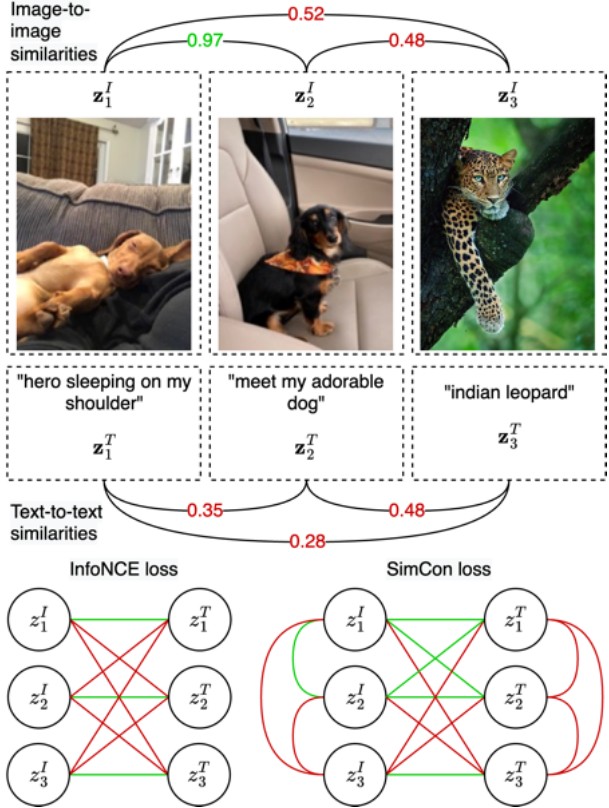

Figure 1: **Top**: In the above image-text samples, the text corresponding to the first image does not contain any information about its visual content. The text for the second image correctly describes the visual content of both the first and the second images. **Bottom**: If we use the InfoNCE loss, only the image and its paired text will be pulled together (green edges), while all other images and captions will be pushed apart (red edges). In SimCon loss, the semantically similar first and second images will be pulled closer to each other and to the text from the second image. The loss function learns this automatically. Best viewed in color.

This work attains state-of-the-art performance for zero-shot text-supervised segmentation on three datasets by mitigating the above-mentioned noisy training issue in the only existing baseline for the task, *i.e.*, GroupViT Xu et al. (2022a). This is accomplished by the demonstrated synergy between the two following elements. *First*, a new loss function termed *SimCon*, which does self-guided semantic alignment by comparing an image jointly with the other image and text samples in the training batch. As shown in Fig. 1, within the batch for an image anchor, other images and their associated texts are assigned as positives if the intra-modal image-to-image similarity is higher than the threshold. The SimCon loss then pulls the embedding vectors of the anchors' corresponding text, positive image samples, and their corresponding texts closer. It pushes apart the embedding vectors of the remaining images and texts. Similarly, the loss for a text samples as an anchor is computed using the text-to-text similarities to determine adequate semantic positives. *Second*, the robustness of contrastive visual representation learning Chen & He (2021); He et al. (2020a); Chen et al. (2021); Caron et al. (2020; 2021); Khosla et al. (2020); Aljundi et al. (2023) is methodically combined with the SimCon loss. Where different views of an anchor image are synthetically generated via data augmentation. This setup is termed *MV-SimCon* for brevity. In the standard training with InfoNCE, the vision model is trained only with supervision from the potentially noisy text from the web. In MV-SimCon, the vision encoder trains with supervision from other semantically similar images, their texts, and the different views of the anchor image, which leads to better performance as demonstrated by training on multiple sources of web data independently and combined.

The contributions of this paper are as follows:

- A novel *SimCon* loss with multiple views is introduced to mitigate the issue of noisy image-text pair training.

- The proposed loss is robust across different data distributions during training and scales with the amount of training data and batch size.

- Extensive empirical results demonstrate the superiority of *MV-SimCon* in terms of faster convergence and better zero-shot segmentation performance.

## 2 Related Work

**Semantic segmentation without dense supervision.** Semantic segmentation is a dense prediction task that assigns a semantic label to each pixel. Most methods rely on annotated data to achieve decent segmentation results Chen et al. (2017a,b); Zhao et al. (2017); Long et al. (2015); Li et al. (2022b); Zheng et al. (2021); Strudel et al. (2021); Cheng et al. (2021); Xie et al. (2021); Zhu et al. (2021). Given costly pixel-wise annotations, there have been several attempts to do unsupervised semantic segmentation. One way is to learn representations for each pixel and then perform clustering to obtain dense semantic labels. Earlier work Ji et al. (2019); Ouali et al. (2020) demonstrates the possibility of clustering on small-scale datasets. To facilitate object discovery, recent work has started to incorporate more prior information Cho et al. (2021); Van Gansbeke et al. (2021); Gansbeke et al. (2022) or better image representations Hamilton et al. (2022); Zadaianchuk et al. (2022); Ziegler & Asano (2022). Yet the performance still lags its supervised counterparts. Another way is to use language supervision as a weak signal. Several recent papers Xu et al. (2022b); Li et al. (2022a); Rao et al. (2022); Lüddecke & Ecker (2022); Zhou et al. (2022) leverage the CLIP model Radford et al. (2021) to enable open-set semantic segmentation, but they also require dense supervision. OpenSeg Ghiasi et al. (2022) goes one step further by using image-level supervision to learn with class-agnostic mask annotation. Finally, GroupViT Xu et al. (2022a) shows that semantic segmentation can be done by training a model on web data image-text pairs without mask annotations. In this work, we instantiate our framework using GroupViT but change the training objectives from InfoNCE to the proposed MV-SimCon. The goal is to mitigate the effect of noisy image-text supervision in GroupViT training.

**Learning visual representations from web data.** Web data as a source of supervision has been a promising direction to learn visual representations Li et al. (2017); Patel et al. (2018; 2019); Gomez et al. (2017); Gordo & Larlus (2017). With the help of metadata like tags and alt-text, the labeling cost of such datasets can be reduced significantly, which leads to cheaper large-scale datasets. In order to study the effect of data in the deep learning era, YFCC100M Thomee et al. (2015), JFT300M Sun et al. (2017), JFT3B Zhai et al. (2022a), IG3.5B Mahajan et al. (2018) and IG65M Ghadiyaram et al. (2019) were collected and studied. As expected, larger datasets help to learn better visual representations and lead to state-of-the-art results for various vision tasks. With the rise of multi-modality learning, there is a recent trend of using image-text pairs from the web as supervision Radford et al. (2021); Jia et al. (2021); Lu et al. (2021); Yu et al. (2022); Wang et al. (2022). Thanks to the larger datasets Schuhmann et al. (2021) and larger transformer models Zhai et al. (2022a); Chen et al. (2022), these trained models have capabilities, such as zero-shot prediction.

**Contrastive loss objectives.** Contrastive losses have been applied to a wide variety of data from several domains, e.g., computer vision Chen & He (2021); He et al. (2020a); Chen et al. (2021); Caron et al. (2020), natural language processing Pan et al. (2021); Gao et al. (2021), speech and audio van den Oord et al. (2018); Saeed et al. (2021) and multi-modal Radford et al. (2021); Jia et al. (2021). These losses can be used as long as the anchor, positives, and negatives are well-defined. Such losses have widely been studied in the open-set image retrieval literature Musgrave et al. (2020), where the deep embedding is trained on a set of classes, and the retrieval evaluation is performed on unseen classes from the same distribution. The simplest pairwise loss function is the contrastive loss Hadsell et al. (2006), also known as InfoNCE, where the embeddings of the relevant pair of samples are pulled as close as possible, and the non-relevant ones are pushed apart. The triplet loss Spyromitros-Xioufis et al. (2014); Weinberger & Saul (2009) mimics a ranking objective more closely by training on a triplet pair of an anchor, a positive and a negative sample. Since optimization over all possible combination of samples is not tractable, much attention has been paid to finding informative

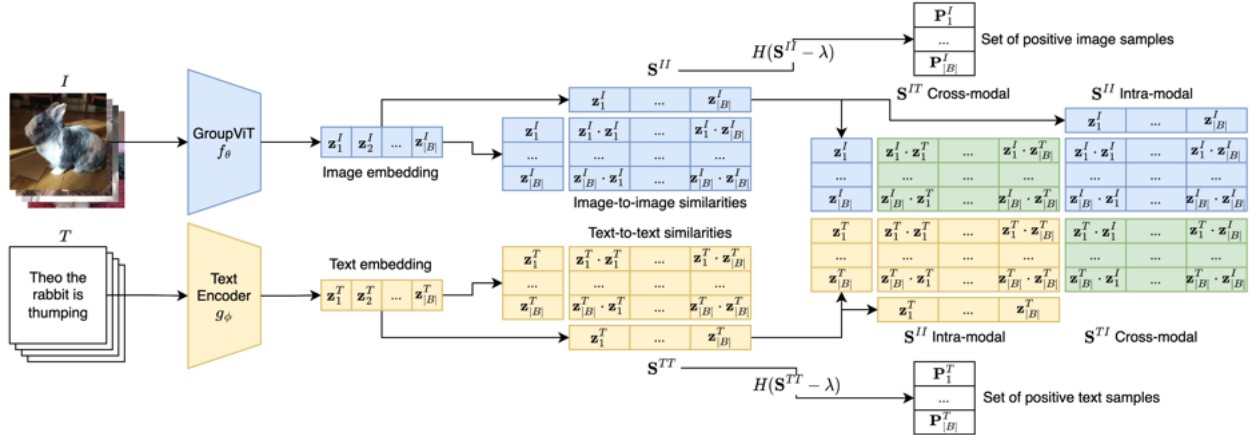

Figure 2: **SimCon Overview.** During training, the sampled images $I$ are passed through the GroupViT model $f_\theta$, and the segment tokens are averaged and normalized to obtain the embedding $\mathbf{z}^I$. The texts $T$ are passed through the text encoder $g_\phi$ to obtain the text embedding $\mathbf{z}^T$. The intra-modal image-to-image $\mathbf{S}^{II}$ and text-to-text $\mathbf{S}^{TT}$ similarities are computed via the cosine distance. The set of positives $\mathbf{P}^I$ and $\mathbf{P}^T$ are determined using the intra-modal similarities by finding the samples with a similarity higher than the threshold $\lambda$. This is achieved by passing $\mathbf{S}^{II} - \lambda$ and $\mathbf{S}^{TT} - \lambda$ through a Heaviside step function $H$, the output 1 determines positives and 0 for negatives. The SimCon loss defined in Eq. (3) and (4) is computed on a joint similarity matrix containing both the intra-modal and cross-modal similarities with the positive and negative relations between the pairs governed by $\mathbf{P}^I$ and $\mathbf{P}^T$. Blue shows image modality, yellow shows text modality and green shows cross-modal. MV-SimCon follows a similar pipeline with multiple views as elaborated in Sec. 3.3.

pairs via sampling Oh Song et al. (2016); Sohn (2016); Roth et al. (2020); Wu et al. (2017); Lu et al. (2019); Brown et al. (2020); Rolínek et al. (2020); Cakir et al. (2019); Patel et al. (2022); Zolfaghari et al. (2021). All the above-mentioned loss functions involving approximation of the evaluation metric or sampling have been studied in a uni-modal, supervised setup with class-balance sampling, which is not feasible with the multi-modal data from the web without any semantic labels. Therefore, the standard InfoNCE Hadsell et al. (2006) has been popularly adopted to date due to its simplicity. Another work, CrossCLR Zolfaghari et al. (2021) uses intra-modal similarities from an expert pre-trained model to prune the set of negatives, a formulation different than ours, where intra-modal similarities are used to find additional positives from the other domain. Our work takes motivation from the image retrieval approaches and attempts to find a better sampling strategy for multi-modal contrastive learning, in the absence of semantic labels and training from scratch. We incorporate intra-modal similarities and multiple views to determine adequate semantically positive samples in the noisy data from the web.

**Vision-language training with noisy data.** Recent approaches, designed mainly for cross-modal retrieval such as ALBEF Li et al. (2021), TCL Yang et al. (2022), CodeBook Duan et al. (2022), and BLIP Li et al. (2022c) also attempt to mitigate noise during training in various ways. While Radford et al. (2021); Jia et al. (2021); Xu et al. (2022a) use separate text and image encoders, Li et al. (2021); Yang et al. (2022); Duan et al. (2022); Li et al. (2022c) also use a multi-modal encoder. For noise mitigation, Li et al. (2021); Yang et al. (2022); Duan et al. (2022) use distillation through an exponentially moving average momentum encoder, whereas Li et al. (2022c) curates the training data using additional captioning and filtering models. These approaches are computationally expensive as they require additional models, have only been investigated for tasks requiring global predictions, such as cross-modal retrieval, where their efficacy is marginal.

# 3 Method

This section revisits GroupViT as a baseline to describe its model architecture and training objectives. Then we introduce our improved contrastive losses by bringing intra-modal similarity (SimCon) (Sec. 3.2) and multi-view (MV-SimCon) into the picture (Sec. 3.3).

## 3.1 Preliminary

**GroupViT.** GroupViT Xu et al. (2022a) is the first work to explore zero-shot transfer from text supervision alone to semantic segmentation without using any pixel-wise labels. The basic idea is to bring back the grouping mechanism Yu et al. (2002); Shi & Malik (1997); Yu & Shi (2001) into deep transformer networks in a bottom-up manner. Through a hierarchical grouping process, the model learns to grow image regions into progressively larger arbitrarily-shaped segments.

To be specific, GroupViT consists of a vision encoder $f_\theta$ and a text encoder $g_\phi$. The vision encoder is a vision transformer (ViT) with group tokens and grouping block. Given a batch of images and the corresponding texts $\{(I_i, T_i)\}$, where $i$ is the index in the batch size, the batch is sampled from a collection of multi-modal data. During feed-forward, the image is first split into non-overlapping patches and linearly projected into a latent space, which are termed as segment tokens. The segment tokens and the learnable grouping tokens are then fed to the transformer layers. After a set of transformer layers, the segment tokens and the grouping tokens are passed to a grouping block. Within the grouping block, segment tokens are assigned to groups and merged together for further processing. The assignment is done by computing the similarities between the segment tokens and the group tokens and using a differentiable Gumbel-softmax assignment Jang et al. (2017); Maddison et al. (2016). The merging combines all the segment tokens belonging to the same group and is performed via a weighted sum. In this way, the group tokens can thus learn to aggregate information globally from all segment tokens. The set of transformer layers followed by a grouping block constitutes a stage. Stacking two such stages stacked together gives the final vision encoder. For training, a global representation of the image $\{\mathbf{z}_i^I\} \in \mathbb{R}^d$ is obtained by average pooling the final segment tokens, followed by $L_2$-norm. The text encoder is a transformer model and is the same as in Radford et al. (2021) and the final normalized text embeddings are denoted as $\{\mathbf{z}_i^T\} \in \mathbb{R}^d$.

**Notations.** Usually the similarity between any two embedding vectors is computed by the dot product between them and is denoted by $s(\mathbf{z}_i^I, \mathbf{z}_i^T) = \mathbf{z}_i^I \cdot \mathbf{z}_i^T$. Within the batch $B = \{(\mathbf{z}_i^I, \mathbf{z}_i^T)\}$, the similarity between all images and texts is computed and is stored in a $|B| \times |B|$ dimensional matrix $\mathbf{S}^{IT}$. For brevity, $\mathbf{S}_{ij}^{IT}$ is the similarity between an image with index $i$ and text with index $j$. Similarly a matrix $\mathbf{S}^{II}$ contains the image-to-image similarities and $\mathbf{S}^{TT}$ text-to-text similarities. The computation of InfoNCE loss involves the temperature controlled exponent of the similarities, and it is represented as $\mathbf{E}_{ij}^{IT} = \exp(\mathbf{z}_i^I \cdot \mathbf{z}_j^T/\tau) = \exp(\mathbf{S}_{ij}^{IT}/\tau)$. Here, $\tau$ is a learnable temperature parameter initialized with a value of 0.07 Radford et al. (2021); Xu et al. (2022a). Similarly the exponent term between two images is represented as $\mathbf{E}_{ij}^{II}$ and between two pieces of text as $\mathbf{E}_{ij}^{TT}$.

**InfoNCE loss.** With a global representation for both image and text modalities, GroupViT uses InfoNCE loss function that matches an image to the corresponding text and vice versa. InfoNCE loss pulls the representations of the corresponding image and text pairs closer and pushes the representations of non-matching (according to data) text samples apart. This image-text alignment loss jointly trains the visual and the textual encoders, and may be expressed for image-to-text matching as:

$$\mathcal{L}_{\text{NCE}}(I, T) = -\frac{1}{|B|} \sum_{i=1}^{|B|} \log \frac{\mathbf{E}_{ii}^{IT}}{\sum_{j=1}^{|B|} \mathbf{E}_{ij}^{IT}} \tag{1}$$

and similarly for text-to-image matching as:

$$\mathcal{L}_{\text{NCE}}(T, I) = -\frac{1}{|B|} \sum_{i=1}^{|B|} \log \frac{\mathbf{E}_{ii}^{TI}}{\sum_{j=1}^{|B|} \mathbf{E}_{ij}^{TI}} \tag{2}$$

where $\mathbf{E}^{TI}$ is the transpose of $\mathbf{E}^{IT}$. The overall training loss of GroupViT is a linear combination of Eq. (1) and (2).

**Shortcomings of InfoNCE.** Given an image and its corresponding aligned text (according to the groundtruth), InfoNCE loss pulls the image embedding closer to its corresponding text embedding and pushes the other text embeddings away. As shown in Fig. 1, this objective falls short in accounting for noise in the training data where the corresponding text does not contain any or partial information about the visual content of the image. Additionally, InfoNCE does not account for any intra-modal relations. The proposed SimCon loss mitigates this issue as an image is not only positively matched to the corresponding text, but also to additional images and texts determined via the intra-modal similarities.

Furthermore, the InfoNCE loss function does not account for any relations across different views of the image which has shown to be useful for visual representation learning Chen & He (2021); He et al. (2020a); Caron et al. (2020); Chen et al. (2021); Caron et al. (2021) and is available without any supervision, as these views are synthetically generated by applying augmentations. To add a further training signal, the SimCon loss is systematically extended to use multiple views of the images, termed MV-SimCon. Where the images in each view are positively matched to the text following the SimCon objective and an additional cosine distance loss Chen & He (2021) is used to match the images from different views.

### 3.2 SimCon: Self-Guided Semantic Alignment

The proposed SimCon loss jointly accounts for intra-modal and cross-modal relations. For an image anchor, the representations of the corresponding text, semantically similar images, and their corresponding texts are pulled closer while the rest of the image and text representations are pushed apart. An overview of the process is shown in Fig. 2. The SimCon loss for image-to-text alignment may be expressed as:

$$\mathcal{L}_{\text{SimCon}}(I, T, \mathbf{P}^I)$$
$$= -\frac{1}{|B|} \sum_{i=1}^{|B|} \frac{1}{|\mathbf{P}_i^I|} \sum_{p \in \mathbf{P}_i^I} \log \frac{\mathbf{E}_{ip}^{IT} + \mathbf{E}_{ip}^{II}}{\sum_{j=1}^{|B|} \mathbf{E}_{ij}^{IT} + \sum_{j=1}^{|B|} \mathbf{E}_{ij}^{II}} \tag{3}$$

and similarly for text-to-image alignment as:

$$\mathcal{L}_{\text{SimCon}}(T, I, \mathbf{P}^T) = -\frac{1}{|B|} \sum_{i=1}^{|B|} \frac{1}{|\mathbf{P}_i^T|} \sum_{p \in \mathbf{P}_i^T} \log \frac{\mathbf{E}_{ip}^{TI} + \mathbf{E}_{ip}^{TT}}{\sum_{j=1}^{|B|} \mathbf{E}_{ij}^{TI} + \sum_{j=1}^{|B|} \mathbf{E}_{ij}^{TT}} \tag{4}$$

Here $\mathbf{P}_i^I$ is the set of images that are similar to the image anchor at $i$ and $\mathbf{P}_i^T$ is the set of texts that are similar to the text anchor at $i$. As shown in Fig. 2, $\mathbf{P}^I$ and $\mathbf{P}^T$ are obtained from the intra-modal similarities as:

$$\mathbf{P}^I = H(\mathbf{S}^{II} - \lambda)$$
$$\mathbf{P}^T = H(\mathbf{S}^{TT} - \lambda) \tag{5}$$

where $H$ is a Heaviside step function with $H(x) = 1$ if $x >= 0$, otherwise $H(x) = 0$, thus the values in $\mathbf{P}^I$ and $\mathbf{P}^T$ are binary. $\lambda$ is a threshold hyper-parameter on the intra-modal similarities. For an image anchor $\mathbf{z}_i^I$, the positive samples are the ones which have the intra-modal similarity higher than the threshold, *i.e.*, where $\mathbf{P}_i^I = 1$.

### 3.3 MV-SimCon: SimCon with Multiple Views

The proposed MV-SimCon loss, or SimCon loss with multiple views, enforces consistency across multiple image views obtained via data augmentation. Let $I_1$ and $I_2$ be the two views of an image, then the MV-SimCon loss for image to text alignment may be expressed as:

$$\mathcal{L}_{\text{MV-SimCon}}(I, T, \mathbf{P}_J^I) = \mathcal{L}_{\text{SimCon}}(I_1, T, \mathbf{P}_J^I) + \mathcal{L}_{\text{SimCon}}(I_2, T, \mathbf{P}_J^I) \tag{6}$$

and for text to image alignment as:

$$\mathcal{L}_{\text{MV-SimCon}}(T, I, \mathbf{P}^T) = \mathcal{L}_{\text{SimCon}}(T, I_1, \mathbf{P}^T) + \mathcal{L}_{\text{SimCon}}(T, I_2, \mathbf{P}^T) \tag{7}$$

where $\mathbf{P}_J^I$ is the set of images that are similar to the anchor image in either view and $\mathbf{P}^T$ remains the same as in SimCon loss as no text augmentations are used. $\mathbf{P}_J^I$ for the MV-SimCon loss may be expressed as:

$$\mathbf{P}_J^I = H(\max(\mathbf{S}^{I_1 I_1}, \mathbf{S}^{I_2 I_2}) - \lambda) \tag{8}$$

where $\mathbf{S}^{I_1 I_1}$ are the intra-modal similarities within the first view of the images and $\mathbf{S}^{I_2 I_2}$ within the second view. Note that there are two possibilities, the first is to independently compute the image positives in each view and the second is to compute them jointly across the views. The joint computation of the image positives leads to more number of positives for each image and empirically leads to a better performance as shown in the ablations Sec. 4.3.

So far, the MV-SimCon loss aligns the images in both views to the appropriate texts following the SimCon loss. However, the images in the two views are still not connected. To connect them to each other and for an additional training signal, a negative cosine similarity loss Chen & He (2021) is used between the two views of the image:

$$\mathcal{L}_{\text{NCS}}(I_1, I_2) = -\frac{1}{|B|} \sum_{i=1}^{|B|} \frac{1}{2} p(\mathbf{z}_i^{I_1}) \cdot \text{sg}(\mathbf{z}_i^{I_2}) + \frac{1}{2} p(\mathbf{z}_i^{I_2}) \cdot \text{sg}(\mathbf{z}_i^{I_1}) \tag{9}$$

where $sg$ is the stop-gradient operation and $p$ is a projection head Chen & He (2021). The overall objective with the MV-SimCon loss is governed by a linear combination of Eq. (6), (7) and (9),

$$\mathcal{L}_{\text{final}} = \mathcal{L}_{\text{MV-SimCon}}(I, T, \mathbf{P}_J^I) + \mathcal{L}_{\text{MV-SimCon}}(T, I, \mathbf{P}^T) + \mathcal{L}_{\text{NCS}}(I_1, I_2) \tag{10}$$

The design choices in MV-SimCon are methodically made based on the empirical evidence as studied in Sec. 4.3.

## 4 Experiments

The proposed SimCon and MV-SimCon objectives are general and can be used for CLIP style of training Radford et al. (2021) on large scale dataset such as LAION Schuhmann et al. (2021). The focus of this paper is on text supervised semantic segmentation for which the experiments are presented in this section. Additional evaluations on zero-shot classification and cross-modal retrieval are presented in the appendix. The implementation details along with clear differences with the baseline Xu et al. (2022a) and a discussion on multi-label loss are in the appendix.

### 4.1 Training and Evaluation Datasets

**Training datasets.** The experiments use Google's Conceptual Captions GCC3M Sharma et al. (2018), GCC12M Changpinyo et al. (2021), RedCaps12M Desai et al. (2021) and filtered YFCC14M Thomee et al.

| Loss Function | Training Data | PASCAL VOC | PASCAL Context | COCO | Average |
|---|---|---|---|---|---|
| InfoNCE Xu et al. (2022a) | CC3M | 16.0 | 7.20 | 6.50 | 9.90 |
| SimCon (Ours) | CC3M | 30.4 +14.4% | 15.1 +7.90% | 12.2 +5.70% | 19.2 +9.30% |
| MV-SimCon (Ours) | CC3M | 35.0 +19.0% | 17.1 +9.90% | 13.4 +6.90% | 21.8 +11.9% |
| InfoNCE Xu et al. (2022a) | R12M | 19.1 | 11.0 | 8.9 | 13.0 |
| SimCon (Ours) | R12M | 37.9 +18.8% | 18.1 +7.10% | 19.5 +10.6% | 25.2 +12.2% |
| MV-SimCon (Ours) | R12M | 40.7 +21.6% | 19.1 +8.10% | 21.6 +12.7% | 27.1 +14.1% |
| InfoNCE Xu et al. (2022a) | CC12M | 41.4 | 19.6 | 20.5 | 27.1 |
| SimCon (Ours) | CC12M | 47.1 +5.70% | 21.3 +1.70% | 22.6 +2.10% | 30.3 +3.20% |
| MV-SimCon (Ours) | CC12M | 48.9 +7.50% | 23.0 +3.40% | 23.8 +3.30% | 31.9 +4.80% |

Table 2: Zero-shot semantic segmentation results on PASCAL-VOC Everingham et al. (2010), PASCAL Context Mottaghi et al. (2014), and COCO Lin et al. (2014) measured with mask mIoU (%) with different training loss functions. Each model is trained independently either on GCC3M Sharma et al. (2018), Red-Caps12M Desai et al. (2021), or GCC12M Changpinyo et al. (2021) dataset with the same setup. Absolute improvements (%) over the baseline Xu et al. (2022a) are shown in blue.

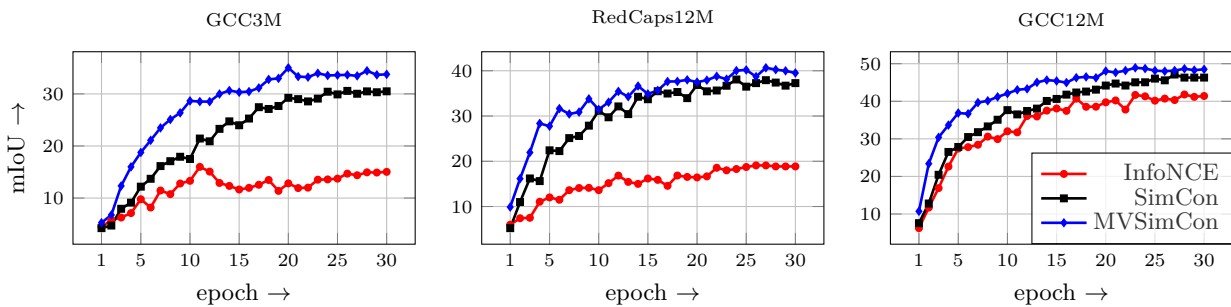

Figure 3: Zero-shot semantic segmentation results on PASCAL VOC Everingham et al. (2010) measured with mask mIoU (%) after each training epoch.

(2015). The exact number of samples for these datasets in our version, along with the number of samples in GroupViT Xu et al. (2022a) implementation, are reported in Tab. 1. Note that the images in these datasets are hosted on a range of sources on the web, where the links change or expire with time. Therefore, the number of samples in our version of the dataset is lower than those in Xu et al. (2022a). To investigate the efficacy of the proposed method, we experiment with different numbers of training samples, starting with 3 million from GCC3M to 41 million with all the mentioned datasets combined. Further, we also experiment with different distributions at a similar scale by comparing models trained on GCC12M with ones trained on RedCaps12M.

| Dataset | Avg. Length | #Samples Ours | #Samples Xu et al. (2022a) | % diff. |
|---|---|---|---|---|
| GCC3M Sharma et al. (2018) | 10.5 | 2.857M | 2.891M | −1.2% |
| GCC12M Changpinyo et al. (2021) | 22.4 | 10.696M | 11.156M | −4.1% |
| Redcaps12M Desai et al. (2021) | 11.8 | 11.835M | 11.866M | −0.3% |
| YFCC14M Thomee et al. (2015) | 38.4 | 14.611M | 14.615M | −.03% |

Table 1: Datasets used for the training along with the number of samples in our and in GroupViT's Xu et al. (2022a) version of the datasets.

| Model | Arch. | Pre-training Dataset | Supervision | Zero shot | TTA | PASCAL VOC | PASCAL Context | COCO |
|---|---|---|---|---|---|---|---|---|
| DeiT Touvron et al. (2021) | ViT | ImageNet (1.2M) | class | ✗ | ✗ | 53.0 | 35.9 | - |
| DINO Caron et al. (2021) | ViT | ImageNet (1.2M) | self | ✗ | ✗ | 39.1 | 20.4 | - |
| DINO Caron et al. (2021) | ViT | CC3 + CC12 + Y14 (29M) | self | ✗ | ✗ | 37.6 | 22.8 | - |
| MoCo He et al. (2020a) | ViT | ImageNet (1.2M) | self | ✗ | ✗ | 34.3 | 21.3 | - |
| MoCo He et al. (2020a) | ViT | CC3 + CC12 + Y14 (29M) | self | ✗ | ✗ | 36.1 | 23.0 | - |
| InfoNCE Xu et al. (2022a) | GroupViT | CC3 + CC12 + R12 (27M) | text | ✓ | ✗ | 50.8 | 23.7 | 27.5 |
| InfoNCE + Multi-label Xu et al. (2022a) | GroupViT | CC3 + CC12 + Y14 (29M) | text | ✓ | ✗ | 52.3 | 22.4 | 24.3 |
| InfoNCE† Xu et al. (2022a) | GroupViT | CC3 + CC12 + R12 (27M) | text | ✓ | ✗ | 44.7 | 20.0 | 23.4 |
| InfoNCE† Xu et al. (2022a) | GroupViT | CC3 + CC12 + Y14 (29M) | text | ✓ | ✗ | 50.3 | 21.7 | 24.6 |
| InfoNCE† Xu et al. (2022a) | GroupViT | CC3 + CC12 + R12 + Y14 (41M) | text | ✓ | ✗ | 50.5 | 20.9 | 23.0 |
| MV-SimCon† | GroupViT | CC3 + CC12 + R12 (27M) | text | ✓ | ✗ | 52.3 +7.60% | 24.5 +4.50% | 27.7 +4.30% |
| MV-SimCon† | GroupViT | CC3 + CC12 + Y14 (29M) | text | ✓ | ✗ | 52.4 +2.10% | 22.2 +0.50% | 26.6 +2.00% |
| MV-SimCon† | GroupViT | CC3 + CC12 + R12 + Y14 (41M) | text | ✓ | ✗ | 53.5 +3.00% | 24.2 +3.30% | 29.9 +6.90% |
| InfoNCE† Xu et al. (2022a) | GroupViT | CC3 + CC12 + R12 + Y14 (41M) | text | ✓ | ✓ | 53.2 | 22.7 | 24.8 |
| MV-SimCon† | GroupViT | CC3 + CC12 + R12 + Y14 (41M) | text | ✓ | ✓ | 58.7 +5.50% | 26.6 +3.90% | 33.3 +8.50% |

Table 3: Mask mIoU (%) on PASCAL-VOC Everingham et al. (2010), PASCAL Context Mottaghi et al. (2014) and COCO Lin et al. (2014) datasets. Comparisons between zero-shot and fully supervised transfer. Zero-shot "✓" indicates transfer to semantic segmentation without any fine-tuning. Absolute improvements (%) over the baseline Xu et al. (2022a) are shown in blue. The authors of this paper train all models that are marked with a † with the same data and batch size. In gray are the models trained by the authors of Xu et al. (2022a) with different versions of the data and a higher batch size. TTA "✓" indicates the use of test time augmentations for inference such as flip, multiple scales, and evaluation at higher resolution.

**Evaluation datasets.** The proposed approach is evaluated for the task of zero-shot transfer to semantic segmentation on the validation sets of PASCAL VOC Everingham et al. (2010), PASCAL Context Mottaghi et al. (2014), and Microsoft COCO Lin et al. (2014). They each contain 20, 59, and 80 foreground classes, respectively, with an additional background class. For COCO, following GroupVIT Xu et al. (2022a), the instance segmentation masks from the same class are combined to obtain segmentation masks.

## 4.2 Evaluation

**Independent training datasets.** The results by training the model independently on either GCC3M Sharma et al. (2018), RedCaps12M Desai et al. (2021) or GCC12M Changpinyo et al. (2021) are shown in Tab. 2, where improvements are observed with the use of every pre-training dataset and across all evaluation datasets. The proposed SimCon loss function shows a substantial improvement ranging from an average gain of 3.2% to 9.3%. The proposed MV-SimCon setup demonstrates additional improvements on top of SimCon. Note that most of the improvements come from using SimCon with MV-SimCon adding an additional 0.4% to 2.6% to the average performance. While InfoNCE achieves an average performance of 27.1% mIoU when training on GCC12M, it only attains an average performance of 13.0% mIoU when training on RedCaps12M, which is of a similar scale in terms of the number of samples. This result shows that training with the InfoNCE loss is highly sensitive to data distribution. On the other hand, the proposed MV-SimCon attains 31.9% mIoU average performance when trained on GCC12M and 27.1% mIoU average performance when trained on RedCaps12M, which demonstrates the robustness of MV-SimCon across different data distributions at the same scale.

The performance on PASCAL VOC Everingham et al. (2010) measured after each training epoch is shown in Fig. 3. MV-SimCon converges faster than SimCon, while both SimCon and MV-SimCon improve and converge faster than InfoNCE. When training on GCC3M, SimCon and MV-SimCon outperform the final performance of InfoNCE (after 30 epochs) after only 7 and 3 epochs of training, respectively. Similar observations were made while training on RedCaps12M. On GCC12M, MV-SimCon improves the fastest.

**Combining training datasets.** Results by training the model on different combinations of training datasets, along with comparisons to fully supervised transfer, are shown in Tab. 3. For these experiments, GCC3M and GCC12M are always used and are combined with either RedCaps12M or YFCC14M or both. Under the same training setup, *i.e.*, the same batch size and training dataset version, the proposed MV-SimCon consistently outperforms InfoNCE loss on all evaluation datasets. Note that increasing the number of training samples from 29 million to 41 million for InfoNCE marginally improves the results on PASCAL

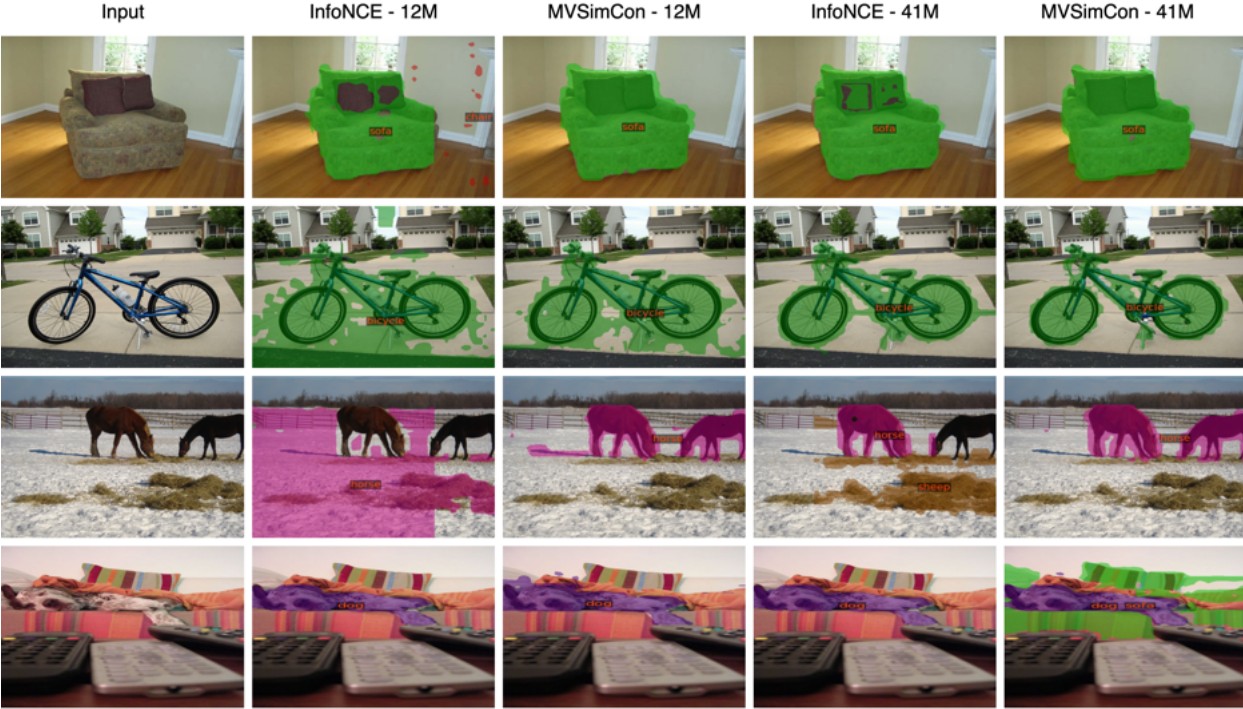

Figure 4: Qualitative results with models trained on 12 and 41 million samples. The 12M setup is trained on GCC12M Changpinyo et al. (2021) dataset, and the 41M setup is trained on a combination of all datasets in Tab. 1. Best viewed in color and by zooming in to see the predicted class.

VOC and degrades the results on PASCAL Context and COCO. Furthermore, the performance on all evaluation datasets is worse, with 27 million in training samples. On the other hand, MV-SimCon is more robust to the domain distribution at a comparable number of training samples, and the improvements hold with an increased number of training samples. In comparison to the models trained by the authors of GroupViT Xu et al. (2022a) with a higher batch size and more training samples in each dataset, the proposed MV-SimCon demonstrates higher scores with smaller batch size and less data. The last two rows on Tab. 3 show results with test time augmentations and higher resolution for inference.

**Qualitative Results.**  Visualization of semantic segmentation predictions from different models is shown in Fig. 4. The models trained with 41 million samples perform better than those trained with 12 million samples for both loss functions. With the same number of training samples, the grouping with MV-SimCon is better and does not have an overly enlarged mask or holes in the mask. Further, the model trained with InfoNCE misses certain semantic classes completely, an issue which is mitigated to some extent with MV-SimCon. More visualizations, including failure cases, are presented in the appendix.

### 4.3   Effect of design choices in MV-SimCon

**Ablation.**  The effect of design choices in MV-SimCon are summarized in Tab. 4. The experiments were conducted by training on GCC3M and evaluating the models on PASCAL VOC. It can be seen that most of the improvements come from using SimCon loss (row-1 vs. row-2). Naively adding multiple views to SimCon, by aligning the images in each view to the text improves the results marginally (row-2 vs. row-3). Adding the negative cosine similarity (NCS) loss between the two views of the image (Eq. (9)) improves the results further by 1.2% (row-3 vs. row-4). Adding joint computation of the image positives (Eq. (8)) improves the results by 2.7% (row-4 vs. row-5). Note that joint image positives refer to the setup when the

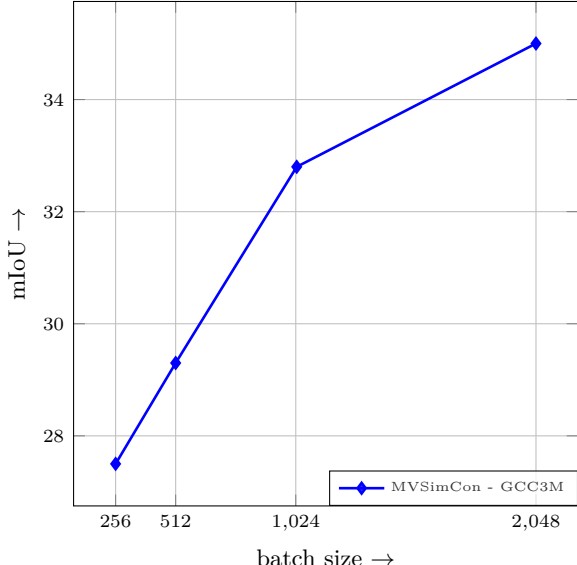

Figure 5: Effect of batch size on GCC3M. x-axis is the batch size and y-axis is the mIoU (in %) on PASCAL VOC dataset.

positive samples for images are assigned if the intra-modal similarity is higher than the threshold in either view.

| SimCon | Multiple views | NCS | Joint image positives | PASCAL VOC |
|:---:|:---:|:---:|:---:|:---:|
| ✗ | ✗ | ✗ | ✗ | 16.0 |
| ✓ | ✗ | ✗ | ✗ | 30.4 |
| ✓ | ✓ | ✗ | ✗ | 31.1 |
| ✓ | ✓ | ✓ | ✗ | 32.3 |
| ✓ | ✓ | ✓ | ✓ | 35.0 |

Table 4: Effect of the design choices in MV-SimCon. All the experiments are conducted by training on GCC3M dataset.

**Effect of batch size.** Results with varying the batch size are shown in Fig. 5. Increasing the batch size improves the results. Our main experiments are with a batch size of 2048, whereas the Xu et al. (2022a) uses 4096. This study justifies the lower results for the baseline Xu et al. (2022a) in our experiments than the ones reported by the authors. Based on the trend in Fig. 5, using a higher batch size will improve performance for both our approach and the baseline. However, all the comparisons made in our experiments are fair and were conducted with the same batch size and data.

## 5   Conclusions

A novel loss function termed SimCon is proposed, where an image (text) sample should not only align to the corresponding text (image) but also with the text from samples that are similar in the visual (textual) space. Training is further made robust by combining the SimCon loss with multiple views in the visual domain. The empirical results demonstrate that the use of the proposed MV-SimCon loss function leads to SOTA results on zero-shot semantic segmentation, along with faster convergence. The proposed approach is shown to be robust across different data training distributions and scales with the amount of data.

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

# A    Implementation Details

**Training.**   Each model is trained on the specified dataset with a batch size of 2048 for 30 epochs with AdamW optimizer Loshchilov & Hutter (2017). An initial learning rate of $4e^{-6}$ is linearly warmed up to a maximum learning rate of $1.6e^{-3}$ in the first 2 epochs. Following warmup, the learning rate is decayed via the cosine schedule Loshchilov & Hutter (2016). All the experiments use 8 NVIDIA-A100 GPUs with 40GB of memory each. The threshold on intra-modal similarity values for determining positive samples in SimCon and MV-SimCon ($\lambda$ in Eq. (5) and Eq. (8) of the main paper) is initialized with 0.95 and is decayed using a step schedule by 0.05 after the 2nd and the 15th epochs.

**Differences with GroupViT.**   As noted in the paper, our version of these web-based datasets have fewer samples. Furthermore, due to hardware constraints, all the experiments in this paper are conducted with a batch size of 2048, whereas GroupViT Xu et al. (2022a) uses a larger batch size of 4096. In vision-language pre-training tasks with contrastive learning, the use of larger batch sizes has been shown to give better results Zhai et al. (2022b). The effect of batch size on the proposed MV-SimCon is studied in the ablation section of the main paper. With these unavoidable differences in the setup, we compare the proposed approach with our reproduction of GroupViT Xu et al. (2022a) results in the exact same setup Xu (2022).

**Discussion on Multi-label loss.**   In addition to InfoNCE loss, GroupViT Xu et al. (2022a) uses a multi-label contrastive loss function for training. Where the nouns are extracted from the text and fed to a randomly sampled prompt template to construct additional text samples. In the multi-label contrastive loss objective, an image should not only align to the text from the data but also with the auxiliary texts. The use of the multi-label loss function increases the computational requirements as the auxiliary texts are passed through the text encoder. In its default configuration, this generated three auxiliary texts for each sample which restricts the training to datasets that contain long enough texts to extract nouns from, eliminating datasets such as RedCaps12M Desai et al. (2021) due to its short average token length as shown in Tab. 1. of the main paper.

In our experiments for the baseline, using multi-label loss increased the computational and GPU memory costs, and either gave similar or worse results. As an example, when trained on GCC12M using InfoNCE loss alone, the model attains mIoU of 41.4% on PASCAL VOC, whereas a model trained with InfoNCE and multi-label loss attains 41.1% (reported in Xu et al. (2022a)). Additionally, when trained on 27 million samples from GCC3M, GCC12M and YFCC14M, the model achieves a mIoU of 50.3% on PASCAL VOC, whereas the performance degrades to 47.5% when the multi-label loss is used. For these reasons, the multi-label loss is not used in our reproduction of the baseline.

# B    Additional Evaluations

## B.1    ImageNet Zero-Shot Classification

While the focus of this work is on improving text supervised semantic segmentation, we additionally evaluate our models based on the GroupViT vision encoder on ImageNet Dong et al. (2009) zero-shot classification. These results are presented in Tab. 5. Note that the semantic segmentation performance of the same models is reported in Tab. 2. of the main paper. These ImageNet classification results are non-conclusive because to understand the effect of MV-SimCon on classification, a CLIP model Radford et al. (2021) needs to be trained, which uses a ViT image encoder and not a GroupViT image encoder. Furthermore, the GroupViT architecture is based on a ViT-S model, whereas in the smallest architecture setting, CLIP uses ViT-B. Therefore, further analysis is needed for zero-shot classification, which is beyond the scope and focus of this paper.

## B.2    Zero-Shot Cross-Modal Retrieval

The trained GroupViT models were evaluated on COCO Lin et al. (2014), and Flickr Plummer et al. (2015) cross-modal retrieval in a zero-shot setting. These evaluations are presented in Tab. Tab. 6 and were conducted by following the standard evaluation protocol, and test splits Li et al. (2021).

| Loss Function | Training Data | ImageNet |
|---|---|---|
| InfoNCE Xu et al. (2022a) | GCC3M | 13.9 |
| SimCon (Ours) | GCC3M | 14.5 |
| MV-SimCon (Ours) | GCC3M | 16.2 |
| InfoNCE Xu et al. (2022a) | RedCaps12M | 31.8 |
| SimCon (Ours) | RedCaps12M | 32.9 |
| MV-SimCon (Ours) | RedCaps12M | 36.3 |
| InfoNCE Xu et al. (2022a) | GCC12M | 33.3 |
| SimCon (Ours) | GCC12M | 32.3 |
| MV-SimCon (Ours) | GCC12M | 34.2 |

Table 5: Top-1 accuracy (in %) for zero-shot ImageNet classification. Each model is trained independently either on GCC3M Sharma et al. (2018), RedCaps12M Desai et al. (2021) or GCC12M Changpinyo et al. (2021) dataset with the same setup.

| Loss Function | Training Data | COCO T2I | COCO I2T | Flickr T2I | Flickr I2T |
|---|---|---|---|---|---|
| InfoNCE | GCC3M | 8.0 | 8.8 | 15.6 | 19.5 |
| MV-SimCon (Ours) | GCC3M | 9.4 | 9.5 | 17.9 | 21.2 |
| InfoNCE | RedCaps12M | 10.3 | 14.6 | 12.4 | 13.2 |
| MV-SimCon (Ours) | RedCaps12M | 12.7 | 18.8 | 14.7 | 17.5 |
| InfoNCE | GCC12M | 21.0 | 25.3 | 39.9 | 46.9 |
| MV-SimCon (Ours) | GCC12M | 22.2 | 30.7 | 40.4 | 52.9 |

Table 6: Recall@1 results for zero-shot cross-modal retrieval. Each model is trained independently either on GCC3M Sharma et al. (2018), RedCaps12M Desai et al. (2021) or GCC12M Changpinyo et al. (2021) dataset with the same setup.

## B.3 Zero-Shot Semantic Segmentation

Additional results for zero-shot semantic segmentation using the proposed MV-SimCon loss are shown in Tab. 7. In the main experiments, we followed the evaluation protocol of GroupViT Xu et al. (2022a) for a fair comparison. Therefore, the zero-shot semantic segmentation evaluations in the main paper were performed with the same resolution. For the longer side of the image, the value was set to a maximum of 2048, and that of the shorter size was set to 448. For the models trained with the proposed MV-SimCon, we observed that setting the shorter side to a maximum value of 512 improves the performance without any major speed loss at the inference.

Furthermore, we experimented with test time augmentations (TTA), such as using multiple scales and flipping at the inference. With TTA, the inference is roughly 4x slower. These inference setups demonstrate improvements over our main experiments, as seen in Tab. 7.

| Model | Pre-training Dataset | Supervision | zero-shot | Inference Res. | TTA | PASCAL VOC | PASCAL Context | COCO |
|---|---|---|---|---|---|---|---|---|
| MV-SimCon† | CC3 + CC12 + R12 (27M) | text | ✓ | (2048, 448) | ✗ | 52.3 | 24.5 | 27.7 |
| MV-SimCon† | CC3 + CC12 + Y14 (29M) | text | ✓ | (2048, 448) | ✗ | 52.4 | 22.2 | 26.6 |
| MV-SimCon† | CC3 + CC12 + R12 + Y14 (41M) | text | ✓ | (2048, 448) | ✗ | 53.5 | 24.2 | 29.9 |
| MV-SimCon† | CC3 + CC12 + R12 (27M) | text | ✓ | (2048, 512) | ✗ | 53.2 | 24.6 | 28.4 |
| MV-SimCon† | CC3 + CC12 + Y14 (29M) | text | ✓ | (2048, 512) | ✗ | 52.8 | 22.5 | 27.1 |
| MV-SimCon† | CC3 + CC12 + R12 + Y14 (41M) | text | ✓ | (2048, 512) | ✗ | 55.0 | 24.7 | 31.0 |
| MV-SimCon† | CC3 + CC12 + R12 (27M) | text | ✓ | (2048, 512) | ✓ | 56.8 | **27.4** | 30.6 |
| MV-SimCon† | CC3 + CC12 + Y14 (29M) | text | ✓ | (2048, 512) | ✓ | 55.3 | 24.3 | 29.6 |
| MV-SimCon† | CC3 + CC12 + R12 + Y14 (41M) | text | ✓ | (2048, 512) | ✓ | **58.7** | 26.6 | **33.3** |

Table 7: Mask mIoU (%) on PASCAL-VOC Everingham et al. (2010), PASCAL Context Mottaghi et al. (2014) and COCO Lin et al. (2014) datasets. All models that are marked with a † are trained by the authors of this paper with same data and batch size. The inference resolution is shown as the maximum size of the longer and the shorter sides of the image. For TTA (text time augmentations), flip and multiple scales are used.

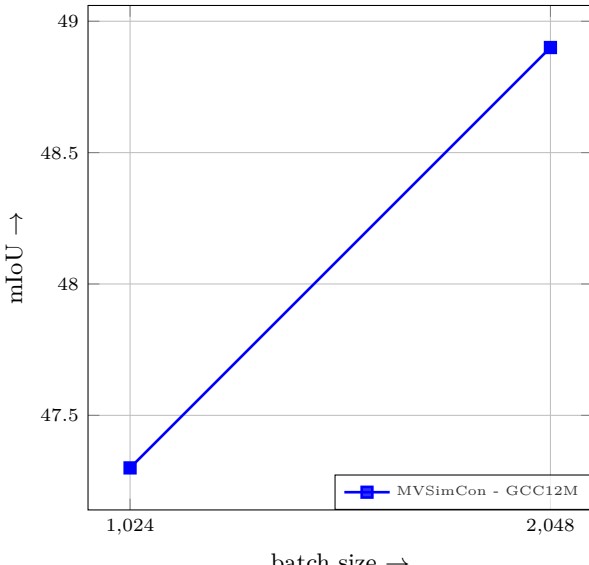

Figure 6: Effect of batch size on GCC12M. x-axis is the batch size and y-axis is the mIoU (in %) on PASCAL VOC dataset.

### B.4 Qualitative Results

We present some failure cases in Fig. 8 where the MV-SimCon model trained on 41 million samples does not give the best results. The description for each shown example is in the caption of the figure.

Additional qualitative results are shown in Fig. 9. The description for each shown example is in the caption of the figure. In most cases, MV-SimCon leads to sharper segments and more accurate recognition than InfoNCE.

## C  Effect of Batch Size on MV-SimCon

Batch size is an important factor for training deep models, particularly for vision-language models using contrastive learning. Here, we study the effect of batch size on the proposed MV-SimCon loss as shown in Fig. 6 (training on CC12M). We can see that increasing the batch size always leads to better performance on zero-shot semantic segmentation. This is consistent with observations in the contrastive learning literature He et al. (2020b); Chen et al. (2020). This potentially explains the difference between our versions of GroupViT and those trained in the original paper (we used a batch size of 2048 dictated by our hardware while the Xu et al. (2022a) used a batch size of 4096).

## D  Visualization of Positives

A visualization of an average number of positives discovered in MV-SimCon training is shown in Fig. 10 and Fig. 11 for 12 million (GCC12M) and 41 million (all datasets) training respectively. We also provide visual samples of additional positives we find during training in Fig. 7 to understand how SimCon works.

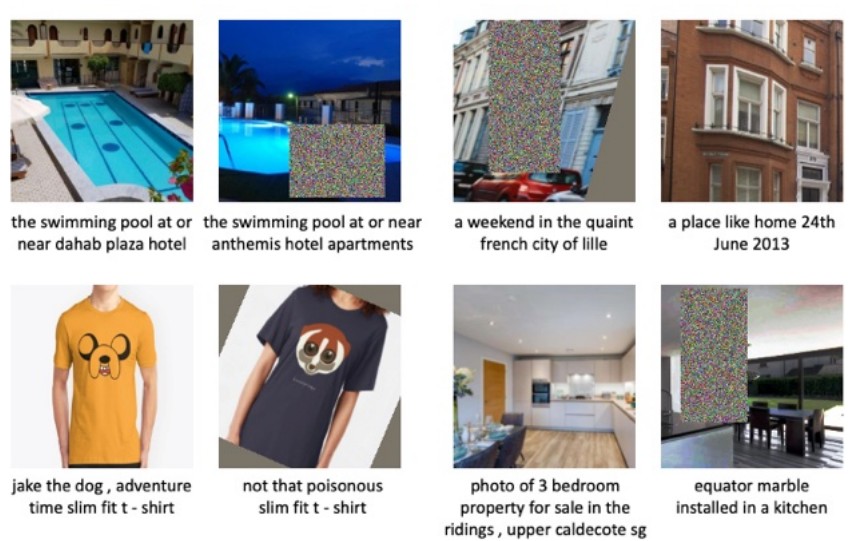

Figure 7: Additional positive samples mined by MV-SimCon during training. Left two examples show that MV-SimCon is able to find similar image-text pairs for richer supervision. The two examples on the right indicate SimCon's capability of noise mitigation, when the original image-text pair is not an exact match. Given these positive samples are selected during training, the images are data augmented.

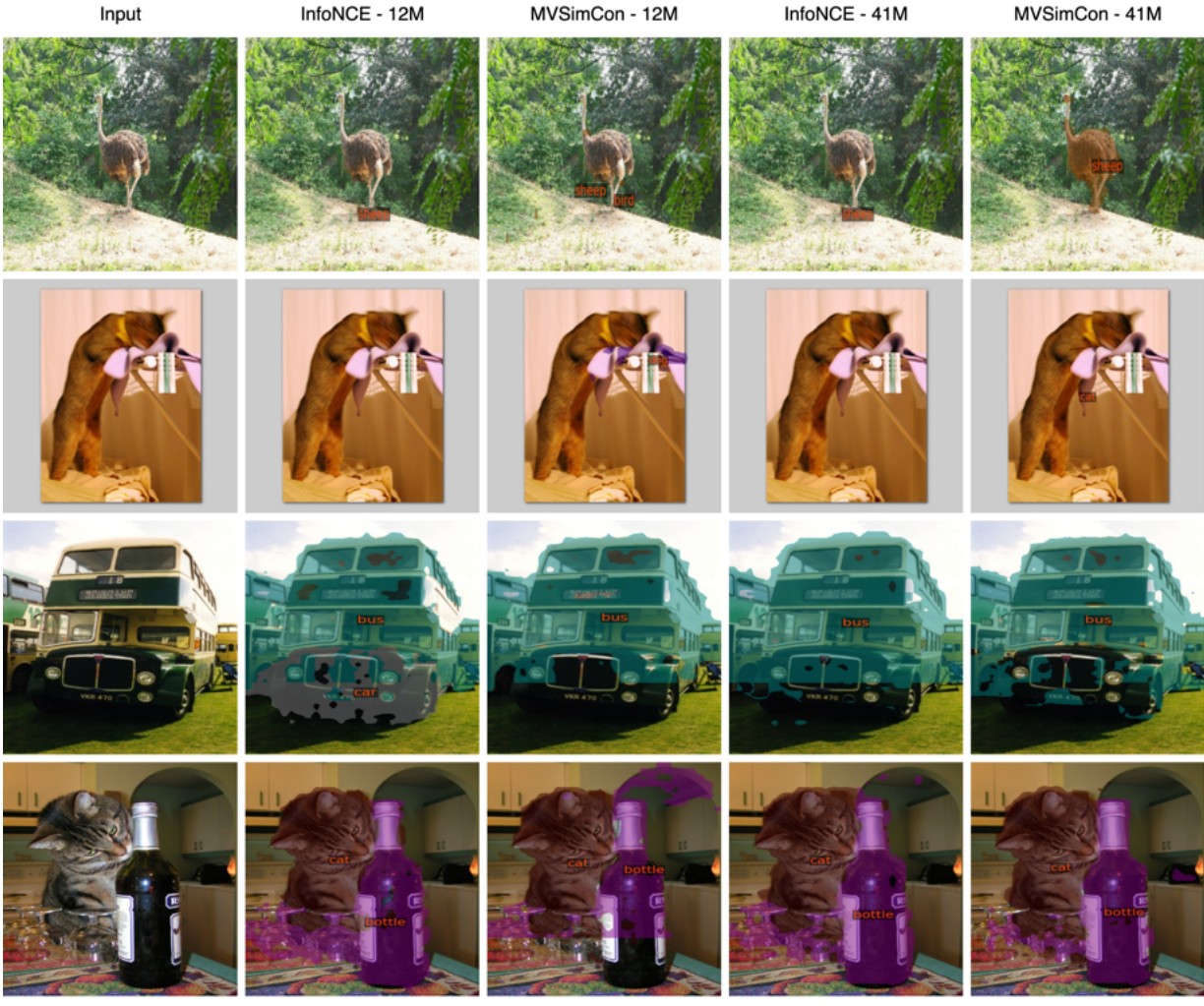

Figure 8: Failure qualitative results with models trained on 12 and 41 million samples. The 12 million setup is trained on GCC12M Changpinyo et al. (2021) dataset and the 41 million setups is trained on a combination of GCC3M Sharma et al. (2018), GCC12M Changpinyo et al. (2021), RedCaps12M Desai et al. (2021) and YFCC14M datasets Thomee et al. (2015). Figure best views in color, the predicted semantic class can be seen by zooming into the figure. In the first example, all the models predict the wrong class label "sheep", only the MVSimCon-12M model predicts the correct class label "bird" (along with "sheep") but with an incorrect semantic mask. In the second example, the InfoNCE based models do not predict any semantic class (entire image is predicted as background). The MVSimCon-41M model predicts the wrong class label "cat" and the MVSimCon-12M model predicts the correct class label "dog". However, the segmentation masks for both MVSimCon models are entirely incorrect. In the third example, all the models fail to segment the bottom part of "bus". InfoNCE-12M model predicts that region incorrectly as "car" whereas rest of the models classify it as background. In fourth and the last example, all the models classify the glasses on the "table" part of the "bottle".

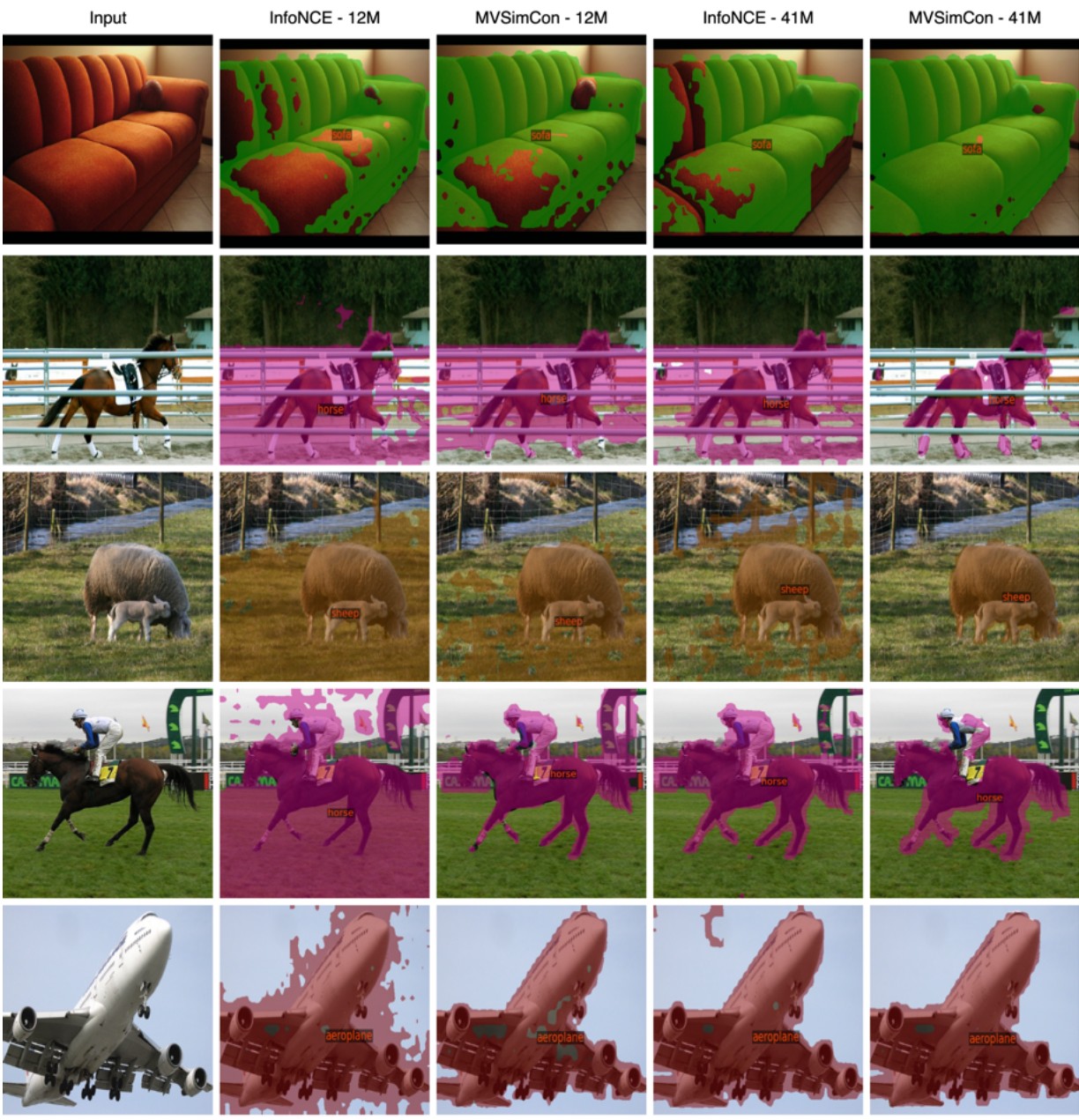

Figure 9: Qualitative results with models trained on 12 and 41 million samples. The 12 million setup is trained on GCC12M Changpinyo et al. (2021) dataset and the 41 million setups is trained on a combination of GCC3M Sharma et al. (2018), GCC12M Changpinyo et al. (2021), RedCaps12M Desai et al. (2021) and YFCC14M datasets Thomee et al. (2015). Figure best views in color, the predicted semantic class can be seen by zooming into the figure. In the first example, all the models predict the semantic class "sofa" correctly with the MVSimCon-41M providing the mask that covers the entire object. In the second example, all the models predict the semantic class "horse" correctly and the MVSimCon-41M model predicts the most accurate mask. The masks predicted by the other models are too large and parts of the background are classified as horse as well. In the third example, all models predict the class "sheep" correctly. However, except the MVSimCon-41M model, rest of the models predict an overly large segmentation mask. In the fourth example, all the models predict the class "horse" but fail to predict the "person" class. For "horse", MVSimCon-41M predicts the most accurate label. In the fifth example, all models correctly predict the class "aeroplane", the InfoNCE models over-segment the object where as the MVSimCon based models predict a more accurate mask.

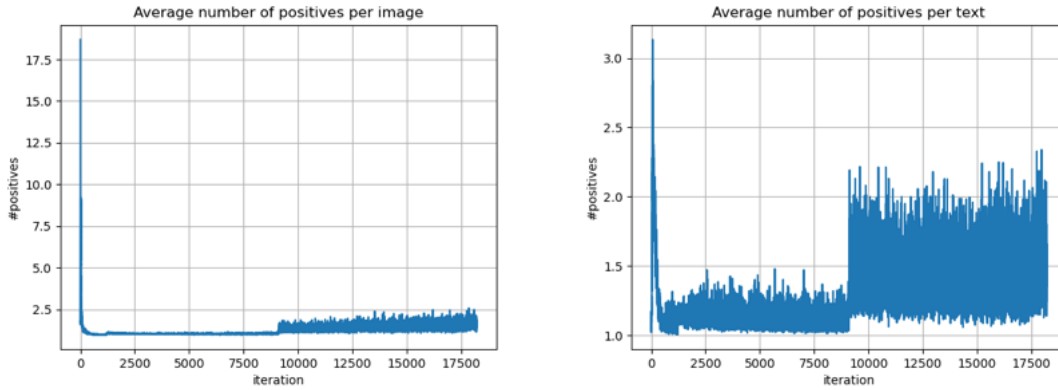

Figure 10: MVSimCon-12M training: Number of positives for images $\mathbf{P}_J^I$ and texts $\mathbf{P}^T$ in MV-SimCon.

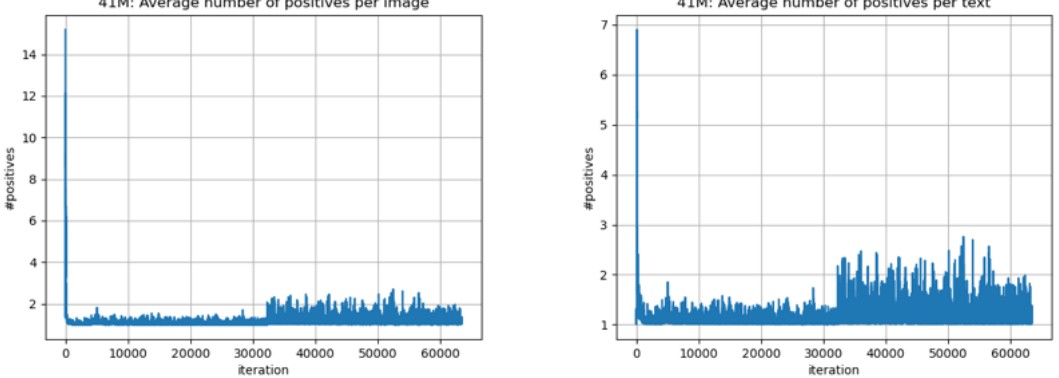

Figure 11: MVSimCon-41M training: Number of positives for images $\mathbf{P}_J^I$ and texts $\mathbf{P}^T$ in MV-SimCon.

