# OpenReview forum: "Self-Guided Semantic Alignment for Text Supervised Segmentation"
_TMLR — Rejected by TMLR_

### Review · Reviewer_Se1W · 2024-03-05

**Summary Of Contributions:**

The authors improve the vision-text feature alignment pipeline usually trained on noisy image-text pairs from the web, by incorporating an intra-modal similarity-weighing scheme for the InfoNCE loss commonly used for training such models. Their formulation of the SimCon loss involves measuring the intra-modal (both image and text) cosine similarity between members of a training batch. This allows representations of images (and text) that are similar to have similar representations despite some image-text pairs being spurious. In a way, this paper completes the thesis of the CLIP model to include self-guidance within each modality. Further, the authors use the representations from a network trained with their modified loss, for zero-shot text-guided segmentation and demonstrate improvement on the task across multiple datasets. They also propose a multi-view loss that considers multiple augmented versions of the same image to compute the image-image similarity for the SimCon loss, likely to smoothen the noise introduced in using the similarity of entirely different images. In the appendix, they also demonstrate improvement in the zero-shot classification and retrieval tasks.

**Audience:**

Yes

**Broader Impact Concerns:**

There are no broader impact concerns.

**Claims And Evidence:**

Yes

**Requested Changes:**

1. Please specify the augmentations used for the multi-view generation, as well as the reason for choosing those.
2. Please specify the value of $\lambda$ indicating whether it is the same for both branches.

**Strengths And Weaknesses:**

Strengths:
1. The paper is very well written, succinctly indicating all the major operations as equations and diagrams.
2. The paper demonstrates state-of-the-art performance for multiple zero-shot segmentation benchmarks and also demonstrates that an improvement is obtained irrespective of the training set.
3. The proposed method also converges faster than the usual InfoNCE-based training.

Weaknesses:
1. The paper fails to adequately attribute its success on the different tasks to the thesis of the paper - which is that the similarity of otherwise misaligned image-text quadruples (i1, t1, i2, t2) is improved with this approach. A test to indicate that (i1, i2) have closer representations than with InfoNCE despite t1 being categorically different than t2, for some number of quadruples could be one way to do that.
2. [Minor] The $\lambda$ used for both intra-modal thresholds appear to be identical. In practice, the similarity scales between text-text and image-image features are significantly different. How do the authors account for this?
3. [Minor] The exact augmentations used to obtain the multiple views of an image are not mentioned in the paper.
4. [Minor] The performance on zero-shot Imagenet Classification (Appendix B table 5)  for SimCon with GCC-12M is worse than that of InfoNCE. This is concerning because it indicates that at the cost of mitigating noise-related issues, the method is losing its discriminative power ever so slightly.
5. [Minor] The authors mention in passing that training with larger datasets like LAION-5B will result in better outcomes. This is unsubstantiated, especially because large datasets typically overcome noise issues in the training data (CLIP etc.).

---

> ### Author Response · Authors · 2024-05-06
> **Response to Se1W**
>
> We thank the reviewer Se1W for the feedback and the time spent on reviewing our submission. We will improve the writing of the paper based on the feedback and remove the unsubstantiated claim regarding training with larger datasets like LAION.
> Response to weaknesses:
>
> 1. **Augmentations**: Our experiments use the same data augmentations as GroupViT, which includes color jitter, auto augment, and random erasing. We will add the details to the paper.
> 2. **Clarification on $\lambda$**: Our experiments used the same value of $\lambda$ for both text-to-text and image-to-image similarities. Overall, we observed that if value of $\lambda$ is too small the training collapses, if the value is too large SimCon does not find enough positives. The choice of $\lambda$ was governed by number of positive samples that were discovered (Fig. 10, Fig. 11 in the appendix), and was kept the same for all experiments at different scales and data distributions. Thank you for the suggestion to use different value of \lambda for the vision and text similarities, intuitively this makes a lot of sense, we leave these investigations for future work.
> 3. **Concerns on zero-shot ImageNet Classification**: Note that the primary focus of this work is on semantic segmentation where the proposed approach has shown consistent improvements across data scale and distribution. For ImageNet classification, we include these results in appendix for completeness. However, we acknowledge that the analysis on classification and retrieval is incomplete as CLIP uses a ViT encoder whereas we use GroupViT encoder. No claims are made in the paper for classification and retrieval.

---

### Review · Reviewer_Lvav · 2024-03-14

**Summary Of Contributions:**

The paper presents a vision-language alignment learning strategy for text-supervised semantic segmentation. To mitigate the noisy supervision signals in InfoNCE, this work develops a loss function consisting of two terms, one of which introduces additional positive pairs based on similar images or texts for infoNCE and the other combines the contrastive loss from visual modality.  The proposed learning framework is evaluated on zero-shot semantic segmentation benchmarks with comparisons to GroupViT.

**Audience:**

Yes

**Broader Impact Concerns:**

No concerns.

**Claims And Evidence:**

No

**Requested Changes:**

1. Improves clarity on the motivation for choosing semantic segmentation.
2. More details on computing the image or text similarities.
3. More complete experimental evaluations as mentioned in Weaknesses.

**Strengths And Weaknesses:**

### Strengths

1. The paper aims to address an important problem in cross-modal learning and proposes a reasonable design for the loss function.
2. It achieves good performance on several semantic segmentation benchmarks compared to GroupViT.

### Weaknesses

1. While the problem of reducing supervision noise in text-image alignment learning is well-motivated, its connection with semantic segmentation is rather weak. The representations are learned at the image level and are quite general. it is unclear why the evaluation has to be performed on image segmentation. Moreover, none of the loss terms are specifically designed for the segmentation problem.

2. The technical novelty of this work is incremental. The proposed loss combines a minor extension of InfoNCE and a commonly-used contrastive loss for visual representation learning. The model architectures are based on ViT or GroupViT.

3. Some of the technical details are unclear. The method relies on a proper similarity between images or texts, which is difficult to obtain without using pre-trained visual or language models. This may lead to unfair comparisons as it uses additional prior knowledge or pre-training.

4. The experimental evaluation is a bit lacking. First, regarding the effectiveness of representation learning, experiments on other downstream tasks should be conducted (as in CLIP). Second, even for the zero-shot image segmentation, the chosen datasets are relatively small in terms of the number of classes. Comparisons on larger-scale benchmarks, such as ADE20k, etc., need to be reported.

---

> ### Author Response · Authors · 2024-05-06
> **Response to Lvav**
>
> We thank the reviewer Lvav for the feedback and the time spent on reviewing our submission.
> Response to weaknesses:
> 1. **Motivation for choosing semantic segmentation**: While the proposed objective functions are generic for CLIP style of models, we choose semantic segmentation primarily because of an established baseline, GroupViT, that trains on tens of millions of scale, which is computationally feasible for us. By default CLIP and OpenCLIP models are trained on hundreds of million (LAION-400M, LAION-CAT) or billion scale samples (LAION-2B), which is computationally not feasible for us. With training on smaller scale, we do report results and comparisons with InfoNCE in the appendix on zero-shot classification and retrieval tasks (Appendix B, Tables 5 and 6).
> 2. **Details on image and text similarities**: We follow the setup for GroupViT and initialize both the vision and the text encoder with random weights. No pre-trained models were used in our setup, and hence the setup is fair. As shown through the empirical results, the proposed method does work despite the randomly initialized encoders. Note that the threshold on intra-modal similarity values for determining positive samples in SimCon and MV-SimCon (in Eq. (5) and Eq. (8) of the main paper) is initialized with $0.95$ and is decayed using a step schedule by $0.05$ after the 2nd and the $15$th epochs. Since the initial value of \lambda is high, the training behaves almost similarly as the InfoNCE for the first two epochs.
> 3. **Additional Results**: Some results for zero-shot classification and cross-modal retrieval are presented in Appendix B Tables 5 & 6. Note that these results are with GroupViT encoder, whereas CLIP uses ViT encoder. On ADE20k, the model trained with InfoNCE loss (baseline) on 41M samples (GCC3M + GCC12M + RedCaps12M + YFCC14M) gets a mIOU of $8.29$% whereas a model trained with MV-SimCon get a mIOU of $11.62$%. A baseline model trained on 27M samples (GCC3M + GCC12M + RedCaps12M) gets a mIOU of $8$%, whereas a model trained with MV-SimCon gets a mIoU of $10.37$%. We will add these results on ADE20k to the main paper.

---

### Review · Reviewer_vUVe · 2024-04-30

**Summary Of Contributions:**

This paper addresses the challenge of noise in training data when learning to segment images based solely on image-text alignment from web sources. The authors propose a new loss function, SimCon, designed to enhance semantic segmentation by comparing images not only to associated texts but also to other images, thereby incorporating intra-modal similarities to identify semantic positives more effectively. An extended version of this loss function, MV-SimCon, leverages multiple synthetic views of an image to further improve. The effectiveness of the method is validated through impressive empirical results on zero-shot semantic segmentation tasks, where they achieve strong performance on the PASCAL VOC, PASCAL Context, and MSCOCO datasets, with additional gains from test time augmentations.

**Audience:**

Yes

**Claims And Evidence:**

No

**Requested Changes:**

Please check weakness for details.

**Strengths And Weaknesses:**

Pro:
1. The proposed idea is intuitive and straightforward.
2. The paper is overall easy to follow.
3. Some results are strong.

Cons:
1. One major concern is the lack of sufficient review of existing work modifying the vanilla InfoNCE objective from CLIP [a,b,c]. The idea of using neighbors [c] or the idea of using more relaxed terms of the loss objective [a,b] are extensively explored in existing literature. Without sufficient review and comparison with existing work, the true position and the contribution of the work cannot be assessed.

2. SOTA is claimed but the compared methods are up till 2022, which are dated to be considered as SOTA. It is also important to properly compare with true SOTA in terms of training cost, inference cost, and accuracy comprehensively.


[a] Robust Cross-Modal Representation Learning with Progressive Self-Distillation
[b] SoftCLIP: Softer Cross-modal Alignment Makes CLIP Stronger
[c] With a Little Help from My Friends: Nearest-Neighbor Contrastive Learning of Visual Representations

---

> ### Author Response · Authors · 2024-05-06
> **Response to vUVe**
>
> We thank the reviewer vUVe for the feedback and the time spent on reviewing our submission.
> Response to weaknesses:
>
> 1. **Review of existing work**: Thank you for pointing out these relevant papers. We will include a discussion on them in the final version of the paper. While the proposed objective functions from these papers add value in the discussion, a direct comparison with them is not feasible as [a,b] build upon CLIP which uses a different vision encoder (ViT) compared to our setup (GroupViT) and [c] reports results for self-supervised visual representation learning, which is a uni-modal task. Note that our task of language guided semantic segmentation is multi-modal and zero-shot, as we do not use any semantic segmentation masks in the training (same as the baseline GroupViT).
> 2. **Claim of state-of-the-art**: We claim state-of-the-art results for text-supervised segmentation without using any segmentation masks or pre-trained vision and text encoders (following the setup of GroupViT). There are other methods since GroupViT that have improved upon the baseline. However, these approaches use pre-trained vision or text encoder from CLIP, trained on substantially larger scale datasets, which makes a fair comparison with them difficult. This includes methods such as SimSeg[a] and CoDe[b]. We will clarify the claim in the paper as what exactly we claim state-of-the-art for and include a discussion on newer paper to make the contributions clearer to the reader.
>     1. Yi et al., A Simple Framework for Text-Supervised Semantic Segmentation, CVPR 2023.
>     2. Wu et al., Image-Text Co-Decomposition for Text-Supervised Semantic Segmentation, arXiv 2024.

---

### Decision · Action_Editor_aNn6 · 2024-07-22

**Recommendation:** Reject

**Comment:**

The paper is reviewed by three expert reviewers.

All three reviewers have concerns about the paper's position, as the link between the proposed loss and the downstream semantic segmentation problem is somewhat weak. Furthermore, Reviewer vUVe has major concerns about insufficient comparison with relevant prior work. Reviewer Se1W commented that "The work has incremental value and has a fundamental problem about the choice of the task."

Regarding the similarity of GroupViT and the proposed work. While both GroupViT focuses on learning semantic segmentation from an image-text dataset, the claimed contributions are different. GroupViT presented a hierarchical Grouping Vision Transformer for this goal. This paper, instead, focuses on improving the generic multi-modal loss function. Thus, this paper should have validated its claimed contributions accordingly (i.e., not as a zero-shot semantic segmentation). The ability to achieve zero-shot segmentation comes from the use of the GroupViT model, not from the proposed method.

In their final recommendations, we have one Reject, one Leaning Reject, and one Learning Accept. As outlined in the Claims and Evidence section, multiple issues remain unresolved. The AE thus has no ground to accept the paper at this time. The authors are welcome to submit a major revision at a later time.

**Audience:**

Yes, the topic of the paper is appropriate to TMLR's audience.

**Claims And Evidence:**

The paper claims new loss functions, SimCon and MV-SimCon, improve zero-shot semantic segmentation performance and SOTA results.

1. Motivation.
- All three reviewers have concerns about the goal of this paper. The proposed modification of InfoNCE loss is at the image level and is not designed explicitly for segmentations. To support the claim of a better generic cross-modal representation, the paper should include all the zero-shot classification, retrieval, and segmentation in the main paper.

2. Comparison with exisitng work on improving InfoNCE loss.
- The claim for improving InfoNCE objective is not well supported without comparisons with existing work that also modified the vanilla InfoNCE objective from CLIP (e.g., the two references provided by Reviewer vUVe). The authors' response is that including that comparison is not feasible due to different feature backbone. Both the reviewer and the AE find it unconvincing since it should be feasible to apply the modified loss to the backbone in the proposed setting. The lack of such results does not substantiate the claimed SOTA performance.

3. Semantic segmentation results.
- As Reviewer vUVe stated, if the paper focuses on semantic segmentation, then more comprehensive evaluations should be performed on various benchmarks and settings (e.g., including pre-trained models).

**Resubmission Of Major Revision:**

The authors may consider submitting a major revision at a later time.